# Targeting the *B1* Gene and Analysis of Its Polymorphism Associated with Awned/Awnless Trait in Russian Germplasm Collections of Common Wheat

**DOI:** 10.3390/plants10112285

**Published:** 2021-10-25

**Authors:** Andrey B. Shcherban, Diana D. Kuvaeva, Olga P. Mitrofanova, Svetlana E. Khverenets, Alexander I. Pryanishnikov, Elena A. Salina

**Affiliations:** 1Institute of Cytology and Genetics SB RAS, Lavrentiev av., 10, 630090 Novosibirsk, Russia; diana.kuvaeva@gmail.com (D.D.K.); salina@bionet.nsc.ru (E.A.S.); 2Kurchatov Genomics Center of ICG SB RAS, Lavrentiev av., 10, 630090 Novosibirsk, Russia; 3NI Vavilov All Russian Inst Plant Genet Resources, 190000 St. Petersburg, Russia; o.mitrofanova@vir.nw.ru; 4Department of Breeding and Seed Production of Agricultural Crops, Schelkovo Agrohim, Zavodskaya Street, 2, 141108 Schelkovo, Russia; mailOE@mail.ru (S.E.K.); a_pryan@mail.ru (A.I.P.)

**Keywords:** ear, cultivar, C2H2 zinc finger, haplotype, marker, *B1* locus, allele, transcription factor

## Abstract

The presence of awns on the ear is associated with a number of important plant properties, such as drought resistance, quality of the grain mass during processing, etc. The main manifestations of this trait are controlled by the *B1* gene, which has recently been identified and encodes the C2H2 zinc finger transcription factor. Based on the previously identified SNPs in the promoter region of this gene, we constructed markers for dominant and recessive alleles which determine awnless and awned phenotypes, respectively. The markers were successful for use in targeting the respective alleles of the *B1* gene in 176 varieties of common wheat, accessions of *T. spelta* L., as well as on F2/F3 hybrids from crosses between awned and awnless forms of *T. aestivum*. We first identified a new allele, *b1mite*, which has both an insert of a miniature Stowaway-like transposon, 261 bp in length, and 33 novel SNPs in the promoter region. Despite these changes, this allele had no effect on the awned phenotype. The possible mechanisms of the influence of the analyzed gene on phenotype are discussed.

## 1. Introduction

Since the introduction of common wheat *Triticum aestivum* L. (2n = 42; genomic formula, BBAADD) into widespread cultivation, it has become obvious that its productivity is largely related to the traits of ear morphology. Intensive selection aimed at increasing yields in various climatic conditions has led to the creation of a large phenotypic diversity of cultivated wheat in terms of traits such as the shape, length, color and hairiness of the ear, awned and awnless, the number of spikelets, grain size, color and shape, etc. One of the important economically valuable traits is the presence of awns on the ear. The awn is a modified leaf blade, which forms a long, pointed appendage on the ear glume. Ear awns effectively increase inflorescence transpiration and play an important role in the processes of photosynthesis and plant respiration [1,2]. About half of the total number of stomata on a spike of wheat are located on the awns [3]. The total surface of the awns captures about 9% of visible solar radiation and increases the efficiency of photosynthesis by 2–12% [4,5]. The positive effect of awns as an additional organ of photosynthesis on quality indicators such as bulk grain weight, 1000 grain weight, flour yield and gluten content has been shown [4,6,7,8,9]. Additionally, the tolerance of awned forms to carbohydrate–protein depletion of seeds has also been established [10,11], as well as their increased resistance to drought and waterlogging [1,12]. Awns also provide protection from animals and promote seed dispersal [13].

With the advent of machine methods of harvesting grain crops, awnless wheat gained a technological advantage over awned wheat, since awns of the latter clog equipment and reduce threshing efficiency, which leads to crop losses. This explains why awnless varieties are much more common in central Russia, the climatic conditions of which do not give a significant advantage to awned forms. However, despite the fact that awnless forms are predominant among domestic spring varieties of common wheat, there is a need to use awned forms in the breeding process, as they are a valuable source of genetic variation, including resistance to abiotic and biotic stress.

For common wheat, it is known that various quantitative indicators of awn presence, awn length and their distribution along the ear are determined by three genetic loci: *Hooded* (*Hd*), *Tipped1* (*B1*) and *Tipped2* (*B2*), located on chromosomes 4A, 5A and 6B, respectively [14,15,16]. Awn presence is most often associated with the *B1* locus in the distal region of chromosome 5AL, which has a strong influence on the phenotypic manifestation of this trait [17,18]. The original recessive allele *b1* controls the awned phenotype in the ancestors of wheat and modern varieties descended from them; the dominant *B1* allele (awn inhibitor) arose as a result of a mutation(s) and was preferred during the selection process. As a result of recent work on genetic mapping of the *B1* locus, it was possible to reduce the region of gene localization to an interval of 0.046 cM, which corresponds to 134 kb of the reference sequence IWGSC RefSeq V1.0 (http://www.wheatgenome.org, accessed on 15 October 2021). In this interval, only two genes were identified, one of which encodes a transcription factor (TF) containing a zinc finger domain (C2H2 zinc finger domain). In this work, based on the data obtained on Chinese common wheat varieties, six SNPs were revealed in the promoter region of the *B1* candidate gene that have a high degree of association with the awned trait [19]. At the moment, enough evidence has been obtained in favor of the fact that this particular gene is responsible for the studied trait [20,21]. However, Huang et al. [20] suggested other polymorphisms in the promoter region of the same gene as mostly predictive of this trait.

The aim of this study was to develop molecular markers for genotyping to discriminate homo- and heterozygotic wheat forms based on awn traits, as well as to test these markers on Russian germplasm collections of common wheat, the closely related species *T. spelta* L. (2n = 42; BBAADD), as well as a population of hybrids obtained from a cross with *T. aestivum*, contrasting in awn traits. Using these markers, the structural polymorphism of the *B1* gene (designated here as *C2H2Zf*) were analyzed and its new alleles associated with the trait under study were studied.

## 2. Results

### 2.1. Analysis of C2H2Zf, Putative Candidate for the B1- Gene

Earlier fine mapping of the *B1* locus identified a single gene that can control awn length. It belongs to the TF family encoding the zinc finger C2H2 domain [19,20,21]. Using the reference sequence of the *T. aestivum* genome, we designed specific primers for PCR amplification and carried out subsequent sequencing of two regions of the specified gene: the coding region, with a length of 363 bp (PCR product—438 bp) and promoter with a length of ~1.2 kb (Appendix A; Appendix A).

Analysis of the primary structure of the coding region of the *C2H2Zf* gene in two awnless cultivars (Start, Saratovskaya 29) and two awned cultivars (Lyubava 5, Element 22) revealed its complete identity in all four cultivars. The analysis of the promoter region was only carried out in the awnless cultivar Saratovskaya 29, and it was then compared to a similar sequence from the cultivar CS (see Plant Material), which carries the recessive allele *b1* [17]. According to the alignment result shown in Appendix A, both varieties differed in six SNPs, with the SNP closest to the gene located at a distance of ~0.7 kb from the start of translation. Next, we carried out a comparative analysis of this gene in awned and awnless accessions from the “10 genomes” database (https://webblast.ipk-gatersleben.de/wheat_ten_genomes/, accessed on 15 October 2021), which contains draft genomic sequences of predominantly European hexaploid wheat varieties. This analysis revealed exactly the same SNP series in four awnless accessions, including three *T. aestivum* cultivars: Julius, Claire, CY Mattis, and one *T. spelta* accession (PI 190962), relative to the other cultivars with an awned phenotype. Therefore, two haplotypes of the *C2H2Zf* gene could be distinguished, which are closely associated with awned and awnless forms of common wheat.

### 2.2. Design and Confirmation of Molecular B1- Markers

Based on the results obtained, we designed and tested various combinations of primers for the *C2H2Zf* promoter region. Combinations of forward and reverse primers annealed within the promoter region gave common, nonspecific amplification products for all cultivars, which are apparently associated with homologous annealing sites in the genomes that are not associated with the studied gene. The best result was obtained by two combinations of forward primers b1for and B1for, which are specific for the haplotypes of awned and awnless varieties, respectively, and the reverse primer Znfrev, which is common to both combinations and anneals at the beginning of the coding region (Appendix A; Appendix A). These combinations were tested on seven spring *T. aestivum* cultivars using the “Touch-down 2” program (Appendix A). As a result, it was confirmed that four awned and three awnless cultivars have corresponding haplotypes and, therefore, these haplotypes are closely linked to the recessive and dominant state of the *B1* locus, respectively.

To further confirm the potential of the developed markers to discriminate awned and awnless genotypes, we performed PCR analysis of material from seven accessions of hexaploid wheat *T. spelta* and five accessions of *T. aestivum*. Of the seven analyzed accessions of *T. spelta*, five gave PCR products with primers B1for/Znfrev and two accessions with primers b1for/Znfrev (Appendix A). Three and two accessions of *T. aestivum* gave PCR products with primers B1for/Znfrev and b1for/Znfrev, respectively.

To assess the phenotype, seeds were planted in a greenhouse and the plants were grown to the heading stage. As in the previous case, we observed correspondence of the phenotypic assessment of the awn trait to the established haplotypes for the *C2H2Zf* gene.

### 2.3. Genotyping and Phenotyping of Russian Wheat Germplasm Collection of VIR

The developed markers were used for genotyping 169 accessions of common winter wheat from the VIR collection, which included 96 accessions from different regions of Russia and 73 accessions from 13 countries of Europe and America (Appendix A).

The presence of awns is one of the traits included in the definition of wheat varieties. The analyzed collection included 169 varieties of common wheat, of which 61 accessions were awned and belonged to varieties *graecum*, *ferrugineum*, *erythrospermum*, *compositumferrugineum* and 108 awnless accessions belonged to varieties *albidum*, *lutescens*, *milturum*, *composituminumilturum velutinum* (Appendix A). The collection material was described previously and tested for awnless/awned traits in the field (Figure 1, Appendix A).

Genotyping of the collection material was carried out using the developed markers for various alleles of the *C2H2Zf* gene: the dominant allele (B1for/Znfrev) and recessive allele (b1for/Znfrev). The analysis results are presented in Figure 2 and Appendix A. No heterozygous forms were found in the VIR collection under study. Using the developed markers, 167 out of 169 accessions were divided into 59 awned and 108 awnless genotypes, which fully corresponded to the description of the varieties presented in the VIR catalog and the results of phenotyping of the accessions in the Oryol region (Russia). The amplification product was absent for two awned accessions (114, 115 Appendix A) of wheat, varieties *graecum* and *compositumferrugineum*. The lack of amplification can be associated with mutations in the primer annealing regions.

Using primer pair b1for/Znfrev, in 11 accessions, the amplification product was about 200 bp larger than the expected product of 1264 bp in length (Figure 2; Appendix A). We isolated the enlarged PCR products from two accessions, No. 2 and No. 13 (Appendix A), as well as the usual PCR product from accession No. 14 as a control, and further analyzed their primary structure. The sequences of the PCR products of the first two accessions were completely identical and differed from the PCR product of the last accession by the presence of a 261 bp miniature transposon (MITE) insertion. In addition to this insertion, located 612 bp upstream the start of translation, 33 SNPs and 1 bp insertion were identified in the promoter region of the *C2H2Zf*- gene (Figure 3). It should be noted that the revealed changes had no effect on the awned trait, since all 11 accessions with the enlarged PCR product were awned, as were the accessions carrying the usual product for the recessive allele *b1*.

### 2.4. Using the B1/b1 Markers for Identification of Heterozygotes and Homozygotes in the Breeding Population

The use of molecular markers for genes associated with an awned or awnless phenotype for the selection of lines from crossing awned and awnless wheat varieties has a number of features, depending on the task at hand. If the breeding task is associated with obtaining awned forms, then the use of *B1/b1* markers is not advisable, since the awned genotype carries the recessive *b1* allele in a homozygous state, and segregation in subsequent generations by the awned trait is not observed. The task becomes more complicated if from crossing awned and awnless genotypes it is supposed to obtain a awnless wheat variety. Here, it is important at the early stages of selection (generation F2) to separate awnless forms into homozygous and heterozygous genotypes for the *B1* gene in order to only select the homozygous *B1B1* genotype for the dominant allele and avoid further segregation into awned and awnless forms in subsequent generations.

In order to test the developed markers for the above breeding task, they were used to select awnless homozygous plants from the population from the crossing of the awned variety Lyubava 5 and the awnless line 476-10 (See Plant Material). Genotyping results for 91 awnless F2 plants using the developed markers for the *C2H2Zf* gene are shown selectively in Figure 4 and in full tabular form in Appendix A.

In total we obtained: 64 plants—heterozygotes (PCR products are present in both combinations) to 27 plants—homozygotes (only PCR product with the B1for/Znfrev combination). Analysis of the F3 generation of the studied genotypes showed that segregation into awned and awnless forms was only observed in the case of 64 *B1b1* heterozygotes identified in the previous generation, while the progeny from F2 awnless plants with the *B1B1* genotype produced the same phenotype in F3 (Appendix A). Thus, the developed markers make it possible to select awnless wheat genotypes in populations from crossing awned and awnless plants, which retain this phenotype in the process of further selection. The use of such markers for the analysis of the F2 generation is allowed to reduce the further volume of the studied material by 2/3.

## 3. Discussion

Work on mapping the main locus controlling the awned trait that has been carried out since 1998 resulted in the approximate localization to the distal part of the long arm of chromosome 5A [17,22,23,24]. Recently, much progress has been made in finely mapping this locus. Almost simultaneously, in three studies, the candidate gene that was most probable to control the awn trait was identified [19,20,21]. This gene encodes the C2H2 Zinc finger transcription factor and belongs to a large family of genes, which have been shown to be involved in plant growth and development, including in the formation of inflorescences and flower organs [25,26]. The gene only contains one exon encoding 121 aa. In this work, we carried out a comparative structural analysis of this gene in various varieties and lines of hexaploid wheat differing in the presence/absence of awns.

Analysis of the primary structure of the coding region of the *C2H2Zf* gene in four Russian awned and awnless cultivars did not reveal any differences. This suggests that the effect of this gene on the manifestation of the studied trait is not related to the structure of its product, but, possibly, is due to the different level of its expression. This is also confirmed by the data of other authors [19,20,21]. The level of expression is known to depend largely on the structure of the promoter region of the gene. We sequenced this region to within ~1.2 kb from the start of translation in the awnless cultivar Saratovskaya 29 (carrier of the dominant *B1* allele) and compared it with a similar region in the CS cultivar, which carries the recessive *b1* allele. As a result, six SNPs were identified, five of which are located at a distance of more than 1 kb from the ATG codon, and one SNP- is located at a distance of ~ 0.7 kb (Appendix A).

The need to develop markers for the awn trait was caused by the fact that currently, there are no unambiguous markers for detecting awned and awnless forms of wheat, including heterozygous awnless genotypes. Previously, different markers were used for this purpose [17,22,23,24], but all of them had different levels of linkage with the trait under study, preventing the reliable differentiation of genotypes, which is only possible with the use of intragenic molecular markers. The identification of the *C2H2Zf* gene, a single probable candidate for *B1*, led to the discovery of a number of polymorphisms upstream of the coding part of this gene [19,20,21]. These polymorphisms include SNPs found by us (see above), together with more distantly located polymorphisms. However, in previous works, there was disagreement about which polymorphisms can claim to be the most effective markers of the trait.

In order to search which SNPs are diagnostic for the trait under study, we carried out a comparative analysis of the *C2H2Zf* gene using draft genomic sequences of seven European *T. aestivum* cultivars and one *T. spelta* accession from the IPK Gatersleben (Germany) database of “10 cultivars”, IPK Gatersleben (Germany). Most of the specimens were represented by awned forms, with the exception of four awnless ones: Julius, Claire, CY Mattis and Spelta. All awnless accessions corresponded to the *C2H2Zf*- gene promoter haplotype characteristic of the Saratovskaya 29 cultivar, while the awned cultivars had the CS haplotype. Thus, six SNPs identified in this and other works within the *C2H2Zf*- gene promoter are reliably associated with awned and awnless phenotypes.

The next stage of the work was the development and testing of markers to identify the recessive and dominant *B1* alleles (*C2H2Zf*). The best discrimination of these alleles was shown by the combination of b1for/Znfrev and B1for/Znfrev primers using the PCR program “Touch-down 2” (Appendix A). In this combination, the forward primers at their 3′-ends contained the two furthest SNPs relative to the gene, while the reverse primer was common and overlapped the beginning of the gene coding sequence (Appendix A). Hence, the genotype was identified as a result of two independent PCR for both alleles, while the presence of only one 1177 bp product in one of the reactions indicates a homozygous state (*b1b1* or *B1B1*), and the presence of products in both reactions indicates a heterozygous state (*B1b1*). The efficiency of using this combination of primers is illustrated by the analysis of the VIR collection, including 96 Russian accessions and 73 accessions from 13 European and American countries (Appendix A). No heterozygous samples were found in the collection under study. The efficiency of identifying dominant and recessive alleles was 100% and 99%, respectively. The high efficiency of the developed markers allowed them to be used for preliminary control of collection material by belonging to the awned and awnless varieties of common wheat at the seedling stage.

Eleven studied accessions of the above collection had an increased PCR product with primers for the identification of recessive allele *b1* (Figure 2). All these accessions represent the Russian germplasm of common wheat from different geographical areas (Appendix A). Five of them belong to subspecies *ferrugineum*; the remaining five (except one with unknown origin) to subspecies *erythrospermum*. Sequencing of this product together with the coding region from two accessions of *erythrospermum* showed that they carry an identical allele with the insertion of a short MITE transposon related to the *Stowaway* family [27] according to its primary structure (Figure 3). BLAST search against the reference sequence of *T. aestivum* (http://plants.ensembl.org/Triticum_aestivum/Info/Index (accessed on 15 October 2021)) revealed 188 sites of localization of sequences with more than 90% homology to the element identified here. It should be noted that the insertion of this element, along with a large number of SNPs (33) in the promoter region of the *C2H2Zf* gene, does not affect the functional state of the gene corresponding to the recessive allele *b1*. From the six nucleotides characteristic of the latter allele (Appendix A), five nucleotides were preserved in the new allele, including those used to construct markers (Figure 3). The coding region contains one SNP that leads to amino acid substitution Alanine (A) → Proline (P). This substitution is located outside the conservative C2H2 zinc finger domain. Taking into account the fact that both amino acids belong to the same chemical class of non-polar amino acids, we suggest that the substitution has no obvious effect on functionality of the protein. The search of the above database “10 genomes” revealed no cultivars with this SNP in their coding regions, as well as the MITE insertion in the promoter region of the *C2H2Zf* gene. Thus, for the first time, we identified a new allele of the *C2H2Zf* gene (*b1mite*) associated with the awned phenotype and that so far, has only been discovered in Russian varieties of common wheat.

The developed primers were also effective in the populations from crossing awned and awnless forms for dividing awnless genotypes carrying the dominant *B1* allele into homozygous and heterozygous genotypes (Figure 4, Appendix A). The removal of heterozygous genotypes from the population significantly reduces the volume of breeding material and avoids further segregation into awned and awnless forms in F3 and later generations. Thus, b1for/Znfrev and B1for/Znfrev primers can be recommended for use in marker-assisted selection during the crossing of awned and awnless cultivars of common wheat.

We analyzed seven accessions of *T. spelta*, one of the oldest species of wheat, cultivated as far back as 7000–8000 BCE in the Neolithic period [28]. According to the results of genotyping, five accessions were homozygous for the dominant *B1* allele, while two accessions had a homozygous genotype for the recessive *b1* allele (Appendix A). These results were completely consistent with the phenotypic analysis, namely the first group showed an awnless phenotype, and the two remaining accessions were awned. The presence of haplotypes characteristic of *T. aestivum* in *T. spelta* can be interpreted in two ways. These haplotypes, corresponding to the awned and awnless forms, could have arisen before the divergence of these species. The second explanation seems more probable: the *B1* awn inhibitor allele could have been introduced into the *T. spelta* genome as a result of natural or artificial hybridization with common wheat. As is known, among the accessions of *T. spelta*, a significant part is made up of accessions more closely related to common wheat according to genotyping data, while true spelt has significant differences from *T. aestivum* [29]. According to N.P. Goncharov, the proportion of awnless specimens in *T. spelta* is approximately 13% [30]. The predominance of awnless forms in the material of the species studied by us is explained by the selectivity of the accessions taken for analysis.

As mentioned above, the *C2H2Zf* gene belongs to a family of regulatory genes with a broad pleiotropic effect on the mechanisms of plant morphogenesis. In particular, a number of members of this family in Arabidopsis regulate the initiation of trichomes through signaling pathways associated with the hormones cytokinin and gibberellin [31]. It is possible that the *C2H2Zf* gene, an awn inhibitor, plays a similar role, controlling the process of awn elongation by suppressing the cytokinin pathway. The effect of cytokinin on cell proliferation is due to the CD25 phosphatase gene, the promoter of which is known to contain binding sites for the C2H2 zinc domain. The established increased level of expression of the dominant allele of the *C2H2Zf* gene correlates with the expression of the *CDC25* gene [19]. In turn, an increased level of synthesis of the *CDC25* gene product can negatively affect the process of stimulation of cell division by cytokinin.

Another important hormone that regulates cell division is auxin. It was shown that the process of formation and division of awn primordium cells is accelerated under the action of auxin [32]. The hypothesis of the influence of the *C2H2Zf* gene on the auxin signaling pathway suggests that an increase in its expression leads to the suppression of the expression of MADS-box-containing proteins, which, in turn, affect the synthesis, transport or perception of auxin [20]. However, the details of this mechanism as well the genes that control it are still unknown.

Long awns can significantly complicate the process of harvesting and processing grain, which can lead to crop losses. However, as an additional organ of photosynthesis, they contribute to an increase in grain weight, especially in drought conditions [4,6,7,8,9]. In addition to enhancing photosynthesis, the *C2H2Zf* gene can affect carbon uptake and also participate in the regulation of the length of glumes, indirectly affecting the grain size [33]. An increase in its expression negatively affects the expression of the *bHLH99* gene in the pericarp at the early stages of grain formation. In turn, this can lead to negative regulation in awnless lines of a number of underlying genes involved in the development of grain tissues [19]. Therefore, silencing or downregulation of the *C2H2Zf* gene by mutations in the promoter part of the gene presents the potential for creating awned lines with a high level of photosynthesis and greater yields through increased grain mass.

## 4. Materials and Methods

### 4.1. Plant Material

The following cultivars of spring common wheat *T. aestivum* from the collection of the ICG SB RAS were used in the work: (1) awned cultivars Lyubava 5, Sintetik, Element 22; (2) awnless cultivars Start, Zauralskaya volna, Saratovskaya 29. The cultivar “Chinese spring” (CS) was used as a control for the recessive allele *b1* (the genomic sequence was deposited in the database “10 genomes” IPK Gatersleben). Additionally, the National Center for Grain (NCG) P.P. Lukyanenko (Krasnodar) provided seeds of common wheat cultivars: Skifyanka, My Vekni, and lines of winter wheat: 4624 k 2-5, 780 k 18-8-11-5, 5303 k 7 (numbering and names of the originator). The analysis also included 169 accessions of *T. aestivum* and 7 accessions of *T. spelta* from the collection of the Vavilov All-Russian Institute of Plant Genetic Resources (VIR) (Appendix A)

The F2 population of common wheat *T. aestivum* was obtained by crossing the awned variety Lyubava 5 and the awnless line 476-10 (early maturing, obtained by crossing the varieties Obskaya and Tulun 15). All awned plants (genotype *b1b1*) were excluded from the population. A list of plants is given in Appendix A. The F3 population obtained by self-pollination of F2 plants was also used in the analysis. Phenotyping of the F2 and F3 populations on the presence of plant awns was carried out under both artificial cultivation and field conditions in the Novosibirsk region, respectively. Phenotyping of the winter wheat collection of VIR was carried out in the Oryol region.

Plant seeds were germinated on moistened filter paper in Petri dishes at room temperature. Genomic DNA was isolated from 7-day-old seedlings according to the method of Kiseleva et al. [34].

### 4.2. PCR

Using the reference sequence of *T. aestivum* genomic DNA RefSeq V1.0, specific primers were designed targeting different regions of the *C2H2Zf* gene (Appendix A). The amplification reaction was carried out on a T100 Thermal Cycler (Bio-Rad, USA) in a volume of 25 μL, containing 2.5 μL of 10x PCR buffer, 2.5 μL of 2.5 mM dNTP mixture, 0.5 μL (1 μM) of forward and reverse primers, 1 unit of *Taq* DNA-polymerase and 50–100 ng of genomic DNA.

Two PCR programs were used: (1) “Touch-down 1” (DNA denaturation, 94 °C—2 min; 12 cycles: 94 °C—45 s, 65 °C—45 s, 72 °C—1 min; 25 cycles: 94 °C—45 s, 55 °C—45 s, 72 °C—1 min; final extension, 72 °C—5 min). (2) “Touch-down 2” (DNA denaturation, 94 °C—2 min; 10 cycles: 94 °C—30 s, 60 °C—30 s (−1 °C each cycle), 72 °C—1 min; 20 cycles: 94 °C—30 s, 50 °C—30 s, 72 °C—1 min; final extension, 72 °C—5 min). PCR products were separated via electrophoresis in a 1% agarose gel containing ethidium bromide in 1xTAE buffer.

### 4.3. Sequencing of PCR Products

PCR products were excised from agarose gels, purified using a QIAquick PCR purification kit (QIAGEN, Germany) and sequenced using a BigDye Terminator v3.1 cycle sequencing kit (Applied Biosystems, USA). Sequencing products were analyzed at the Collective Use Center “Genomika” of the SB RAS. Comparative analysis of DNA sequences was performed using the CLUSTAL O (https://www.ebi.ac.uk/Tools/msa/clustalo/ (accessed on 15 October 2021)).

## 5. Conclusions

Based on the previous studies of the *C2H2Zf* gene (*B1* gene) and our structural analysis of this gene in Russian common wheat cultivars, we first developed specific combinations of primers that effectively discriminate between recessive and dominant *B1* alleles in both homo- and heterozygous states. This has been convincingly demonstrated across a variety of plant materials, including various varieties of spring and winter bread wheat, inter-varietal hybrids of different generations, as well as *T. spelta* specimens from different growing regions. In the future, these combinations of primers can be used in marker-oriented selection based on the awned trait, as well as other important associated traits (grain weight, stress resistance), which are subject to the pleiotropic action of the locus under study. Using the designed primers, we first identified a new allele of the *C2H2Zf* gene containing an insertion of a short MITE transposon together with a number of SNPs. All these changes, however, do not affect the functional state of this gene that leads to the awned phenotype. Except polymorphisms with no visible effect on the phenotype, we found five nucleotide positions preserved in the awned haplotype. It is not yet clear how the identified SNPs in the promoter of the *C2H2Zf* gene affect its expression and what the mechanism of action of this gene is. Nevertheless, a number of indirect and direct data from various authors indicate the involvement of the product of this gene in the hormonal regulation of the growth and proliferation of the primordial cells of the awns.

## Figures and Tables

**Figure 1 plants-10-02285-f001:**
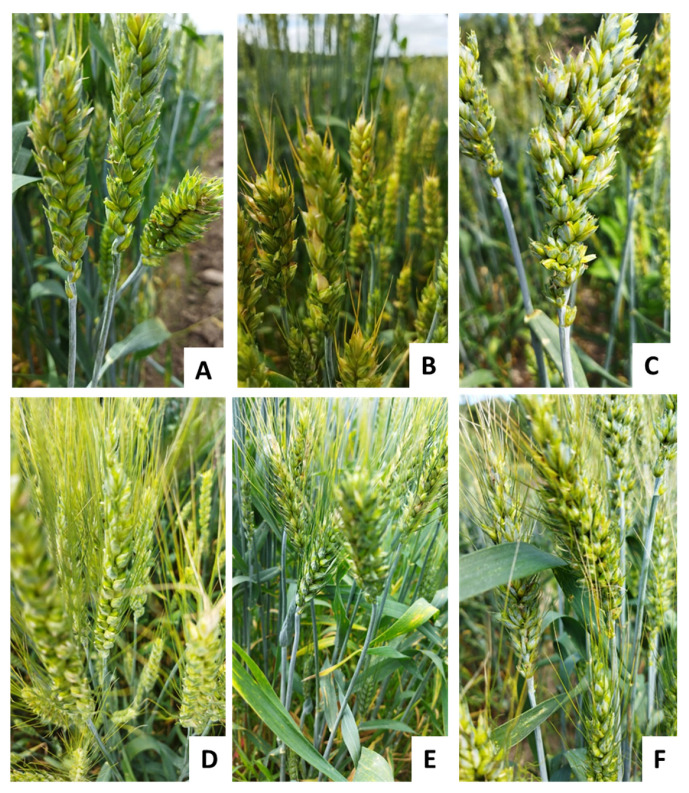
Varieties of awnless (**A**–**C**) and awned (**D**–**F**) accessions of Russian wheat germplasm collection of VIR. (**A**) Voronegskaya 174; (**B**) Donskaya bezostaya; (**C**) Status; (**D**) Voronegskaya 34; (**E**) Labinka; (**F**) Siluet.

**Figure 2 plants-10-02285-f002:**
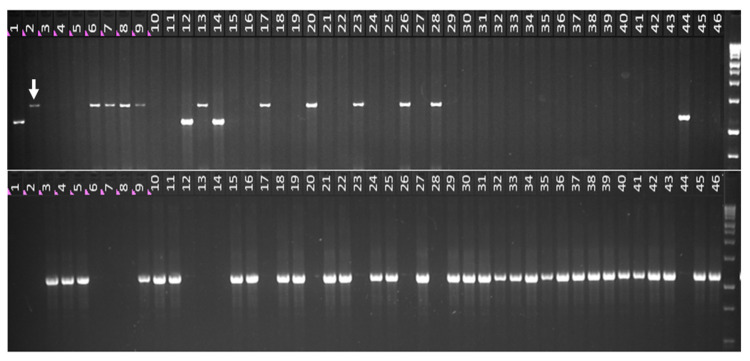
PCR amplification: upper—primers for the recessive allele (b1for/Znfrev); lower—primers for the dominant allele (B1for/Znfrev). For plant numbering, see Appendix A. The arrow indicates an enlarged PCR product with insertion of MITE.

**Figure 3 plants-10-02285-f003:**
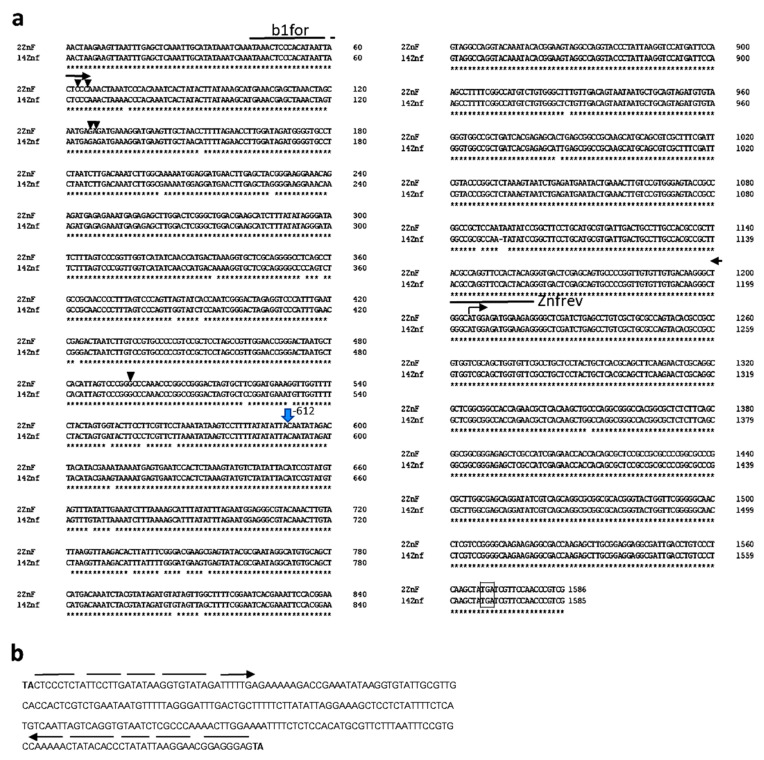
Alignment of the *C2H2Zf*- gene from accessions No. 2 and No. 14 (**a**). An arrow with numbers on the right indicates location of the MITE insertion within the sequence of the accession No. 2. Long arrows indicate position of primers for targeting of dominant and recessive alleles of *B1*. Short vertical arrows indicate 5 SNPs characteristic of awnless trait. Start codon is indicated by the right-angled arrow. Termination codon is framed. The nucleotide sequence of the MITE is presented separately (**b**) with indications of the target site duplication (in bold) and non-ideal flanking inverted repeats (dashed arrows).

**Figure 4 plants-10-02285-f004:**
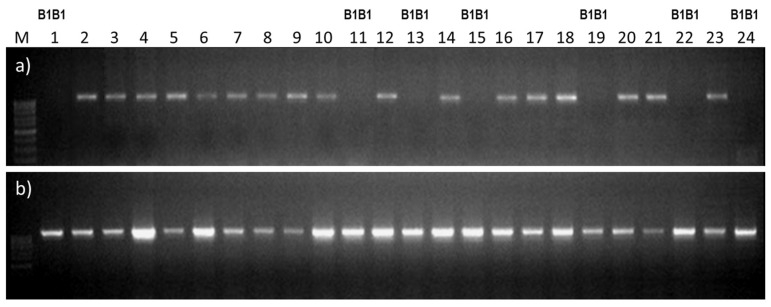
PCR amplification: (**a**) with primers for the recessive allele (b1for/Znfrev); (**b**) with primers for the dominant allele (B1for/Znfrev). For plant numbering, see Appendix A from 1054-1 to 1055-10 inclusively. Homozygous *B1B1* plants are indicated above.

## Data Availability

Data available within the article.

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
