# Peer review of "Targeting the B1 Gene and Analysis of Its Polymorphism Associated with Awned/Awnless Trait in Russian Germplasm Collections of Common Wheat"

_plants, 2021, doi:10.3390/plants10112285_

Round 1
Reviewer 1 Report
I advise a rejection based on the following major concerns,
- As I know, the Chinese Spring (CS) with the HdHdb1b1B2B2 is awnless genotype. But in this manuscript, two haplotypes of the C2H2Zf gene promoter are based on the difference between Saratovskaya 29 and Chinese Spring initially.
- Dominant PCR marker is very inefficient, especially when you need to genotype lots of samples. A KASP marker is more suitable for this study.
Author Response
1.As I know, the Chinese Spring (CS) with the HdHdb1b1B2B2 is awnless genotype. But in this manuscript, two haplotypes of the C2H2Zf gene promoter are based on the difference between Saratovskaya 29 and Chinese Spring initially.
The awness trait is determined not only by the B1 gene, but also by two other genes (Hd and B2), but to a lesser extent than the first. Chinese Spring genotype is HdHdb1b1B2B2 (see ref. 17: Yoshioka M. et al; Three dominant awnless genes in common wheat: Fine mapping, interaction and contribution to diversity in awn shape and length. Plos one 2017, 12(4), e0176148). Two any dominant genes determine awnless phenotype.
We analyzed promotor regions of dominant B1 alleles of S29 and recessive b1 alleles of CS.
Some text correction were made in 2.1.
2.Dominant PCR marker is very inefficient, especially when you need to genotype lots of samples. A KASP marker is more suitable for this study.
Efficiency of the developed markers was shown by the genotyping of a lot of samples. The efficiency for dominant and recessive alleles was 100% and 99%, respectively, for collection comprising 169 accessions of different origin. These are high enough numbers.
Reproducibility of published KASP markers varies and depends on the region of SNP localization and the analyzed genotypes. Besides, we have no information about efficient KASP markers for dominant and recessive alleles which determine awnless and awned phenotypes.
Reviewer 2 Report
Recently three articles reporting gene governing Awnless trait in wheat have been published.
1) DeWitt et al. (2019) Sequence-based mapping identifies a candidate transcription repressor underlying awn suppression at the B1 locus in wheat. New Phytol., 225, pp. 326-339
2) Huang, et al. (2019) Dominant inhibition of awn development by a putative zinc-finger transcriptional repressor expressed at the B1 locus in wheat. New Phytol., 225, pp. 340-355
3) Wang, et al. (2019) Natural variation in the promoter of Awn Length Inhibitor 1 (ALI-1) is associated with awn elongation and grain length in common wheat, Plant J., 101 (2020), pp. 1075-1090
Considering these three papers authors need to restructure the present MS.
Starting from the title, now the b1 gene (TraesCS5A01G542800) is now functionally characterized, and therefore no need to call it a candidate. Considering b1 as a candidate and showing its association with the Awnless trait is reinventing the wheel.
As per the above comment, the author can make their article focused on allele mining in the Russian germplasm.
Figure 1 - this is too superficial. I suggest moving figures 1, 2, and Table 1 to supplementary.
Move figures 3 and 4 to supplementary.
Combine figures 6, and 8 together as a compound figure with multiple panels.
Where is figure 7.
Author Response
Starting from the title, now the b1 gene (TraesCS5A01G542800) is now functionally characterized, and therefore no need to call it a candidate. Considering b1 as a candidate and showing its association with the Awnless trait is reinventing the wheel.
As per the above comment, the author can make their article focused on allele mining in the Russian germplasm.
In accordance with the recommendation, we have changed all the places in the text where the studied gene was considered as a candidate gene. Also, significant changes were made to the title of the article. The focus of the article is shifted from the confirmation of the candidate gene to the development of diagnostic markers based on the already identified gene and the analysis of its polymorphism.
Figure 1 - this is too superficial. I suggest moving figures 1, 2, and Table 1 to supplementary.
Move figures 3 and 4 to supplementary.
Figure 1, 2, 3, 4 and Table 1 have been moved to supplementary.
Combine figures 6, and 8 together as a compound figure with multiple panels.
We would prefer do not combine figures 6 and 8, since they refer to completely different stages of work. Fig 6 (now Fig. 2)- analysis of VIR collection; Fig 8 (now Fig 4)- analysis of F2 generation. But if the editor decides that the 4 main Figures are too many, we could combine them.
Where is figure 7.
Figure 7 now is Figure 3. It was modified. The sequence of b1mite allele now includes not only promoter region but also the recently sequenced complete coding region and upstream part of the promoter with SNPs chatacteristic of the awned phenotype.
Reviewer 3 Report
The manuscript has improved and provides a useful confirmation of the candidate gene controlling the awn phenotype
- Figure 7 is missing so renumber the Figures and make sure they are consistent in the text
- Please provide all the headers for columns in Table S1 in English
Author Response
Figure 7 is missing so renumber the Figures and make sure they are consistent in the text
Figure 7 now is Figure 3 (see the answer to the previous referee).
Please provide all the headers for columns in Table S1 in English
In old version of the table S1 the header “Subspecies” was in Russian. We fixed it.
Reviewer 4 Report
The following are my thoughts on the manuscript are as below:
The abstract must contain information about the author's most significant discoveries, please make it more useful.
The opening must contain a clearly stated hypothesis, and the second paragraph of this article must be extensively expanded on that theory.
A common theme is the repeating of information that may have been overlooked the first time around.
Examine the ligands in the illustration; they have been written on the page at random.
When discussing significant and related works, there should be a greater amount of material and references included. Authors should add other significant studies in the manuscript so that their findings can be compared.
Restructure and carefully revise the conclusion section, with a particular focus on the final paragraph of the section.
Author Response
The abstract must contain information about the author's most significant discoveries, please make it more useful.
We corrected abstract. Now it includes more clear information about our significant discoveries (see answer to referee 1).
The opening must contain a clearly stated hypothesis, and the second paragraph of this article must be extensively expanded on that theory.
The main hypothesis in the old version was if we consider the C2H2Znf gene as a candidate gene for B1, so could we use its polymorphisms (based on previous and our own data) to create effective markers for the studied trait? But now the main focus of the article is not confirmation of the candidate gene, but the search for polymorphisms within the already confirmed (by other authors) gene, which are reliably associated with variability in the studied trait. This is reflected in the introduction (lines 97-103). The rest of the paper in our opinion corresponds to what you call "extensively expanded this theory". First, we have confirmed the effectiveness of the developed markers on a wide range of material. The main achievement in this regard is the identification of at least two SNPs in the promoter, which can be efficiently used in MAS, as well as for further study of the effect of mutations in the promoter of this gene on the studied trait. The second achievement is the discovery of a completely new allele, which, despite numerous changes in the promoter, including the insertion of a mobile element, retained the functionality in determining the awned phenotype. This allele was completely sequenced including the promoter and coding regions.
A common theme is the repeating of information that may have been overlooked the first time around.
All repeating information was removed during reformatting the text.
Examine the ligands in the illustration; they have been written on the page at random.
Thank you, we check
When discussing significant and related works, there should be a greater amount of material and references included. Authors should add other significant studies in the manuscript so that their findings can be compared.
We tried to bring in the discussion all the works related to the topic of gene identification and targeting of the trait under study. The new edition contains additional information about the new allele, obtained at the very last moment, in particular, on the structure of the promoter and coding region. The discussed possible mechanisms of action of this gene are currently exhaustive (nothing new has appeared yet). In our opinion, the number of references is not insufficient (34).
Restructure and carefully revise the conclusion section, with a particular focus on the final paragraph of the section.
We carefully revised the Conclusion so that it clear reflects the most significant results.
Round 2
Reviewer 2 Report
Authors have improved the MS and can be accepted
English editing is required
Reviewer 4 Report
Authors have improved the manuscript therefore it can be accepted for publication.
This manuscript is a resubmission of an earlier submission. The following is a list of the peer review reports and author responses from that submission.
Round 1
Reviewer 1 Report
The manuscript “Structural analysis of a candidate gene controlling awn presence in common wheat Triticum aestivum L.” has very important informations to wheat research community. The development of molecular markers associated to awn trait will be very useful in wheat breeding program. However, the text needs some improvement. A better discussion of the results is necessary.
I wrote some suggestions to improve the manuscript:
Keywords : delete the words awn, gene, common wheat because they are in the title
Please write out numbers under 10.
You should show the F3 results to confirm the rate 2:1. How many awned plants did you obtain in F2 population? How many plants did you get in F2 population ? And in F3 population?
Describe the characterization of presence and absence of awns. Show some pictures if necessary.
Discussion needs to be deeper, with a comparison of this study with other articles, including about the development of the molecular markers associated to important wheat traits.
Did you observe if all awnless genotypes have the same SNPs?
Why did you use 1% agarose gel instead of acrilamide gel?
Reviewer 2 Report
The manuscript could be reconsidered as a review if the journal considers this category of manuscript.
As a research paper the paper it could be possible if reduced to 50% with a simple message relating to defining the SNP markers promoter for the gene controlling the awn phenotype. If the authors used the various genome modification resources available and obtained clear evidence that "the gene" has been identified, the paper would take on a completely different level of significance.
Reviewer 3 Report
In 2020, three papers The Plant Journal (2020) 101, 1075–1090, New Phytologist (2020) 225: 340–355 and New Phytologist (2020) 225: 326–339 identified the awn inhibition B1 locus in wheat encoding a C2H2 zinc finger protein. In the first paper, polymorphisms of four single nucleotide polymorphisms located in promoter region are diagnostic for the B1/b1 genotypes. Is there any difference between the polymorphism in this manuscript and published papers? The reference 19 in this manuscript had been published on the plant Journal https://onlinelibrary.wiley.com/doi/epdf/10.1111/tpj.14575.
The aim of this study claimed by the authors was to find polymorphisms of this gene in awned and awnless varieties of common wheat in Russia. However, they only sequenced 4 cultivars’ coding region and 1 cultivars’ promoter region. The developed molecular markers just used to genotype 12 wheat cultivars and 7 accessions of T. spelta. The studied cultivars were so limited, and the dominant PCR marker was of low inefficiency. When they genotyped the F2 populations, it’s very strange that only the awnless plants were used and the F3 phenotypes were not included. I advise a rejection.
Reviewer 4 Report
Abstract and introduction:
- The abstract must include data regarding the critical finds by the authors with some data related to their findings.
- The introduction must have a clear hypothesis and significantly develop the second paragraph of this manuscript.
Results:
Clearly cite Table and Figures in the results section and elaborate the results wherever required.
Discussion:
Discussion should include more information and references related to the relevant and related works. Check the English in the Discussion section.